# FedSR: A Simple and Effective Domain Generalization Method for Federated Learning

**A. Tuan Nguyen**
University of Oxford
tuan@robots.ox.ac.uk

**Philip H. S. Torr**
University of Oxford
philip.torr@eng.ox.ac.uk

**Ser-Nam Lim**
Meta AI Research
sernamlim@fb.com

## Abstract

Federated Learning (FL) refers to the decentralized and privacy-preserving machine learning framework in which multiple clients collaborate (with the help of a central server) to train a global model without sharing their data. However, most existing FL methods only focus on maximizing the model's performance on the source clients' data (e.g., mobile users) without considering its generalization ability to unknown target data (e.g., a new user). In this paper, we incorporate the problem of Domain Generalization (DG) into Federated Learning to tackle the aforementioned issue. However, virtually all existing DG methods require a centralized setting where data is shared across the domains, which violates the principles of decentralized FL and hence not applicable. To this end, we propose a simple yet novel representation learning framework, namely FedSR, which enables domain generalization while still respecting the decentralized and privacy-preserving natures of this FL setting. Motivated by classical machine learning algorithms, we aim to learn a **simple representation** of the data for better generalization. In particular, we enforce an L2-norm regularizer on the representation and a conditional mutual information (between the representation and the data given the label) regularizer to encourage the model to only learn essential information (while ignoring spurious correlations such as the background). Furthermore, we provide theoretical connections between the above two objectives and representation alignment in domain generalization. Extensive experimental results suggest that our method significantly outperforms relevant baselines in this particular problem.

## 1 Introduction

In this paper, we are interested the problem of decentralized Federated Learning (FL), where $K$ clients collaborate to jointly train a machine learning model with their local (decentralized) data. For privacy reasons, we consider the setting where it is desirable/preferred that the clients do not share their data with each other. For example, [28] brings up an example scenario where mobile devices collaborate to train a model without sending their data to a central server. Another example is that K hospitals train a diagnosis model together but do not want to share their patients' data.

However, a major concern with this machine learning framework is that it often does not account for the possible distribution shift on the target data, a problem commonly referred to as Domain Generalization (DG). Furthermore, and perhaps more problematically, it not straightforward to incorporate existing DG techniques into this FL setting due to its privacy-preserving nature. Specifically, most domain invariant representation learning methods [9, 38, 31, 22, 32] require sharing/comparing the representation of the data across domains, while meta-learning based methods [21, 5] require access to data from a pool of multiple source domains (in a central server).

There have been some early attempts to incorporate domain generalization into the FL framework. For example, [27] proposes a technique tailored for medical image segmentation which allows the

36th Conference on Neural Information Processing Systems (NeurIPS 2022).

clients to share their data in the frequency space with each other. This can be viewed as a form of data leakage and is undesirable in the FL setting we consider. Meanwhile, [44] aligns the representation distribution across domains via a reference distribution (from a generative model). While this does fully preserve the data privacy, it can be overly complicated to implement in practice, and the performance is not yet on par with SOTA centralized methods. Nevertheless, this is one of our main baselines in this paper.

To tackle the above challenges, in this paper, we introduce a simple yet effective method for domain generalization in the FL setting. We propose to learn a "simple" representation of the data; specifically, we employ two regularization methods, namely the L2 regularizer on the representation, and the conditional mutual information between the data and the representation given the label. These regularizers are easy to implement and do not require the sharing of data between clients, thus fully respecting the principles of FL. In particular, they aim to restrict the amount of information the representation can contains, which we hope will help to ignore spurious correlations and lead to better generalization to unseen target domains. With a deeper look, we show that these regularizers act as **implicit** forms of marginal and conditional representation alignment , which also further explains the performance gain of our method. This means that our method attempts to learn a representation whose marginal distribution and conditional distribution (given the label) are invariant across the domains without explicitly comparing those distributions between clients. Our contributions in this work are as follows:

- We propose a simple method, which aims to learn a simple representation of the data for the prediction task, for DG in the FL setting. To this end, we revisit two common regularization methods, namely the conditional mutual information regularizer and the L2-norm regularizer on the representation, to restrict the amount of information it can contain.
- Furthermore, we show theoretically and empirically that these two regularizer are well connected with marginal and conditional alignment in the invariant representation learning framework.
- We validate the effectiveness of our resulting model with a wide range of datasets and model architectures and show that it significantly outperforms existing FL methods on the domain generalization task and can even compete with state-of-the-art centralized methods.

## 2 Related Work

### 2.1 Federated Learning

One of the first and most common FL algorithms is FedAVG [28], which uses several local SGD updates within each communication round to the server to save communication cost. In the context of DG in the FL setting, this method is equivalent to Empirical Risk Minimization in a centralized setting, and is one of our main baselines. One of the main challenges that FL needs to tackle is the statistical diversity among the clients (i.e., each client has a different local data distribution). A great amount of research [23, 47, 12, 24, 17, 36] has been done to tackle this problem (i.e., how to train a shared model to deal with non-i.i.d. data of the source clients in the FL setting). Another line of approaches to deal with this problem is to allow each client to have a personalized model and jointly train the models in a FL way (personalized FL) [13, 6, 4, 8, 16]. Although these above methods tackle the distribution shift across the source domains/clients, they still have not dealt with the generalization ability to an unseen target domain – the problem we consider in this paper.

### 2.2 Domain Generalization

Most of the existing DG methods consider a centralized setting. A predominant and effective approach is to learn a domain-invariant representation [30, 25, 26, 42, 2, 15, 46, 1, 22, 38, 35] (meaning to learn a representation whose marginal distribution and/or conditional distribution given the label are unchanged across the domains), hoping that this representation generalizes better to an unseen target domain. Our method can be viewed as implicit forms of representation alignment, thus more suitable for the FL setting. Another line of approaches for DG is to use the idea of meta-learning [7, 5, 21]; with the logic being that if a model can adapt among the source domains well, it is more likely to adapt to an unseen target domain. These methods typically require access to a pool of data from source domains (in a centralized server). [45] considers a single-domain domain generalization

setting. However, it needs to generate a "fake" target distribution from a source distribution and then maximize the mutual information between the representations of the two domains (source vs fake target). This might be overly complicated when applied at each client level in a FL setting and hence less practical. [34] considers a federated domain adaptation, where each source domain aligns the representation distribution with the target domain by receiving the representations from this target distribution. This cannot be applied to our problem setting to align the representation of two clients, since one client cannot have access to another source client's data (including the data representation) for privacy reasons.

# 3  Approach

## 3.1  Preliminaries

We first present the problem of Federated Learning and the importance of Domain Generalization in this setting.

**Federated Learning**  In the conventional FL setting, $K$ clients collaborate with each other to train a global model. Each clients $i$ ($1 \leq i \leq K$) has its own data distribution $p_i(x, y)$, where $x \in \mathcal{X}$ is the input and $y \in \mathcal{Y}$ is its corresponding label, and a dataset with $N_i$ data points: $\mathcal{D}_i = \{(x_i^{(1)}, y_i^{(1)}), ..., (x_i^{(N_i)}, y_i^{(N_i)})\}$. It is often assumed that the data distribution $p_i(x, y)$ changes across the clients (which we will use interchangeably with domains in this paper). Here, the data distributions $p_i(x, y)$ are sampled from a family $\mathcal{E}$ of distributions ($\mathcal{E}$ can be finite or infinite), i.e., $p_i(x, y) \sim \mathcal{E}$. Our aim is for the clients to jointly train a global model with parameters $w$ (this model will be specified later), with the loss function of a datapoint $(x, y)$ denoted as $\ell(w; x, y)$.

The global objective function to minimize is:

$$f(w) := \frac{1}{K} \sum_{i=1}^{K} f_i(w) \tag{1}$$

where $f_i(w)$ is the local objective function, $f_i(w) = \mathbb{E}_{p_i(x,y)}[\ell(w; x, y)] \approx \frac{1}{N_i} \sum_{n=1}^{N_i} \ell(w; x_i^{(n)}, y_i^{(n)})$. Note that the objective in Eq. 1 can also be a weighted average among the clients. However, since the analysis and discussion are unchanged, we will use this simple average objective in this paper.

In most Federated Learning algorithms, the optimization process consists of a number of rounds. Within each round, each (active) client runs its own local minimization of the local objective for several iterations of stochastic gradient descent; then the central server will collect the newly-learned parameters from all (active) clients, average them to form the new global parameter, and broadcast it back to each client at the beginning of next round. By taking multiple steps of the local minimization each round, the algorithm can reduce the communication cost (which is often the bottleneck). For a more detailed description of these FL methods, we refer the readers to [28, 41, 37].

**Domain Generalization**  In the DG problem, we are more interested in training a model to perform well in an unseen target domain $p_T(x, y) \sim \mathcal{E}$ (which is different from $p_i(x, y) \ \forall i$). Therefore, we hope to minimize not only the expected loss on the source domains/clients $\mathbb{E}_{p_i(x,y)}[\ell(w; x, y)]$ but also the loss on unseen target domains, in either an average or worst-case sense as defined below.

In the average case, we want to minimize this loss over the distribution family $\mathcal{E}$:

$$\mathbb{E}_{p_T \sim \mathcal{E}} \left[ \mathbb{E}_{p_T(x,y)}[\ell(w; x, y)] \right] \tag{2}$$

Meanwhile, we might also be interested in the worst-case quantity, i.e., the loss of the domain in which the model performs the worst:

$$\sup_{p_T \in \mathcal{E}} \left[ \mathbb{E}_{p_T(x,y)}[\ell(w; x, y)] \right] \tag{3}$$

In a conventional machine learning problem, we usually need to observe more **datapoints** to learn to generalize better to **unseen datapoints**. Similarly, in this Domain Generalization problem, we additionally need to observe more **domains** in order to generalize better to **unseen target**

**domains**. However, we typically have a limited number of source domains in these DG problems, so regularization techniques often need to be applied for a better generalization. One common technique among those is to learn a domain invariant representation, which means to learn a representation $z$ of $x$ that has the same marginal and/or conditional distribution across the domains. These methods often help to ignore domain-specific spurious correlations and lead to better generalization. We will discuss this in more detail later in the paper.

**Generalization Challenge of FL**   From the objective in Eq. 1, it is clear that conventional FL methods only focus on maximizing the performance (minimizing the loss) on $K$ source clients' data, which is very problematic in many applications. For example, if $K$ clinical institutes in the US and UK collaborate to train a Covid prediction model, the goal is not only to perform well on their own data distributions but also to generalize well to data from other countries. Similarly, if we use Federated Learning to train a machine learning model with data collected from multiple mobile phone users, it is crucial that the model can generalize to a new user. Thus, domain generalization is very important for Federated Learning.

However, as mentioned earlier, applying existing domain generalization techniques to Federated Learning is **not straightforward**. This is because most existing DG methods need a centralized training setting with data (or a part of the data, such as a representation) being shared across the domains. For example, the aforementioned domain invariant representation learning approach typically requires sharing and comparing the representation distribution across domains. Towards this end, we propose a simple but effective domain invariant representation learning technique for Federated Learning that fully keeps its decentralized and privacy aspects. We first introduce the problem setting of representation learning in Subsection 3.2, and then present our domain invariant representation learning methods in Subsection 3.3.

## 3.2   Problem Setting: Representation Learning for Domain Generalization

In the representation learning framework, we aim to learn a representation $z$ of $x$ with a distribution $p(z|x)$ parameterized by $w_1$, which we omit here for notation simplicity. This can be a deterministic mapping (i.e., $p(z|x) = \delta_{g_{w_1}(x)}(z)$ where $\delta$ is the dirac delta function) or a probabilistic mapping (e.g., $p(z|x) = \mathcal{N}(z; \mu_{w_1}(x), \sigma^2_{w_1}(x))$ where $\mathcal{N}$ is the normal distribution). With the data distribution $p_i(x, y)$ of client $i$, its joint distribution of $x, y, z$ is:

$$p_i(x, y, z) = p_i(x, y)p(z|x), \quad \text{since } z \text{ is conditionally independent of } y \text{ given } x \qquad (4)$$

From the representation $z$, we learn a classifier/regressor that predicts $y$ given $z$ with the predictive distribution $\hat{p}(y|z)$ parameterized by $w_2$ (again, we omit $w_2$ here for notation simplicity). For example, with a classification problem, this is often just a linear layer followed by a softmax layer to form the predictive distribution.

The predictive distribution of $y$ given $x$ of our model is:

$$\hat{p}(y|x) = \mathbb{E}_{p(z|x)}[\hat{p}(y|z)] \qquad (5)$$

Note that when $p(z|x)$ is a deterministic mapping, Eq. 5 simplifies into $\hat{p}(y|z = g_{w_1}(x))$.

For both regression and classification, the loss function of a data point $x, y$ is often the negative log predictive, i.e, $\ell(w; x, y) = -\log \mathbb{E}_{p(z|x)}[\hat{p}(y|z)]$, and in this case $w = \{w_1, w_2\}$.

Hence, the local objective of client $i$ is:

$$f_i(w) = \mathbb{E}_{p_i(x,y)} \left[ -\log \mathbb{E}_{p(z|x)}[\hat{p}(y|z)] \right] \qquad (6)$$

During training, for a probabilistic mapping, we often sample a single $z$ per $x$ from $p(z|x)$ (for a deterministic mapping, it is obvious that we only sample one $z$), leading to the training objective:

$$\overline{f_i}(w) = \mathbb{E}_{p_i(x,y)} \left[ \mathbb{E}_{p(z|x)}[-\log \hat{p}(y|z)] \right] \qquad (7)$$

$$\approx \frac{1}{N_i} \sum_{n=1}^{N_i} -\log \hat{p}(y_i^{(n)}|z_i^{(n)}), \text{ where } z_i^{(n)} \text{ is a single sample from } p(z|x_i^{(n)}) \qquad (8)$$

The quantity $-\log \hat{p}(y|z)$ is often non-negative for all common predictive distributions. For example, with a categorical predictive distribution in a classification problem, this is the cross-entropy loss; while with a Gaussian predictive distribution (with a fixed variance) in a regression problem, this becomes the squared error (with an additive constant).

Note that according to Jensen's Inequality, $\overline{f_i}(w)$ is an upper bound of $f_i(w)$, so the training process is sound.

The representation learning framework that we discussed so far still has not dealt with the problem of domain generalization. Many existing works in DG focus on learning a domain-invariant representation $z$, which means to learn a representation $z$ which has the same marginal ($p_i(z)$) and/or conditional ($p_i(z|y)$ or $p_i(y|z)$) distributions across the domains. However, this often requires sharing information of the data (the representation) across domains/clients, which is not allowed in Federated Learning.

Motivated by the fact that classical machine learning algorithms make predictions based on a simple representation of the data and often generalize better than modern deep learning models, in this paper, we aim to learn a **S**imple **R**epresentation of the data (hence the name FedSR). We apply several well-known regularization techniques to restrict the representation's complexity and encourage the deep network $p(z|x)$ to only learn the essential information. Furthermore, we show that these regularization techniques have strong theoretical connections to domain-invariant representation learning and that they can achieve good generalization performance in practice.

### 3.3 FedSR

As mentioned earlier, we employ an L2-norm regularizer ($\ell_i^{L2R}$) on the representation and (an upper bound of) the conditional mutual information between the data and the representation given the label ($\ell_i^{CMI}$) to restrict the amount of information the representation can contain. Moreover, $\ell_i^{L2R}$ also has the effect to align the marginal distribution $p_i(z)$ to be centered around 0 as a Gaussian distribution, while $\ell_i^{CMI}$ aligns the conditional distribution $p_i(z|y)$ to a reference distribution (also chosen to be Gaussian in the experiment section). These two terms will be explained in detail below. The final local objective function of each client $i$ is:

$$\overline{f_i} + \alpha^{L2R}\ell_i^{L2R} + \alpha^{CMI}\ell_i^{CMI} \tag{9}$$

where $\alpha^{L2R}$ and $\alpha^{CMI}$ are hyper-parameters. (We will also test the two regularizers separately, namely the FedL2R and FedCMI variants.)

#### 3.3.1 L2-norm Regularizer

In optimization and learning theory, L2-norm of the weights/parameters is often regarded to as a measure of the model's complexity. Here, with a similar idea, we enforce an L2-norm Regularizer (L2R) on the representation (not the network parameters) to restrict the complexity of the representation. (Note that some methods like [29] use the L2-norm but in the parameter space to help with the generalization of Federated Learning). Specifically, the regularizer has the form:

$$\ell_i^{L2R} = \mathbb{E}_{p_i(x)}\left[\mathbb{E}_{p(z|x)}[||z||_2^2]\right] \tag{10}$$

$$\approx \frac{1}{N_i}\sum_{n=1}^{N_i}||z_i^{(n)}||_2^2, \text{ where } z_i^{(n)} \text{ is a single sample from } p(z|x_i^{(n)}) \tag{11}$$

Intuitively, this regularizer will have the effect of encouraging $z$ to be centered around 0 for all clients, thus helping with the marginal alignment of the representation. This connection can be explained in more detail as below:

**Connection to Domain Invariant Representation Learning:** First of all, rewrite $\ell_i^{L2R}$ as:

$$\ell_i^{L2R} = \mathbb{E}_{p_i(x)}\left[\mathbb{E}_{p(z|x)}[||z||_2^2]\right] = \mathbb{E}_{p_i(x,z)}\left[||z||_2^2\right] = \mathbb{E}_{p_i(z)}\left[||z||_2^2\right] \tag{12}$$

Now, consider a "reference" distribution $q(z) = \mathcal{N}(0, \sigma^2 I)$ (with a small $\sigma$). Then we have:

$$-\log q(z) = \frac{||z||_2^2}{2\sigma^2} \text{ (with an additive constant)} \tag{13}$$

And hence:

$$\ell_i^{L2R} = \mathbb{E}_{p_i(z)}\left[||z||_2^2\right] = 2\sigma^2 \mathbb{E}_{p_i(z)}\left[-\log q(z)\right] = 2\sigma^2 H(p_i(z), q(z)) \tag{14}$$

where $H(p_i(z), q(z))$ denotes the cross entropy from $q(z)$ to $p_i(z)$.

Note further that:

$$H(p_i(z), q(z)) = H(p_i(z)) + \mathrm{KL}[p_i(z)|q(z)] \tag{15}$$

Hence, if the entropy $H(p_i(z))$ does not change significantly during training, minimizing $\ell_i^{L2R}$ will also minimize $\mathrm{KL}[p_i(z)|q(z)]$, which encourages $p_i(z)$ to be close to $q(z)$, i.e., an implicit alignment of the marginal distribution.

Nevertheless, also according to the above equation, if we really want to minimize $\mathrm{KL}[p_i(z)|q(z)]$ (for the marginal alignment), it is perhaps more rigorous to minimize the regularizer $\ell_i^{L2R} - 2\sigma^2 H(p_i(z))$ instead of $\ell_i^{L2R}$. However, when $\sigma$ is small enough, this term is dominated by $\ell_i^{L2R}$ (meaning that if $\sigma$ is small, minimizing $\ell_i^{L2R}$ almost guarantees to minimize $\mathrm{KL}[p_i(z)|q(z)]$). Empirically, we found that the two variants achieve almost identical performance. We therefore recommend using $\ell_i^{L2R}$ instead of $\ell_i^{L2R} - 2\sigma^2 H(p_i(z))$ for its simplicity.

### 3.3.2 Conditional Mutual Information regularizer

The mutual information term between the input $x$ and the representation $z$ (denoted $I(x, z)$) is often used to regularize the amount of information $z$ can contains [3, 39]. However, this regularizer might be too restrictive in practice if its coefficient is not tuned properly, as it encourages the representation to contain no information about the input data. In this paper, we use the **conditional mutual information** of $x$ and $z$ given $y$, as it is less restricted than the former. We can also later see that this is well-connected with conditional distribution alignment in DG. Recall that this conditional mutual information term is calculated for domain/client $i$ as:

$$I_i(x, z|y) = \mathbb{E}_{p_i(x,y,z)}\left[\log \frac{p_i(x, z|y)}{p_i(x|y)p_i(z|y)}\right] \tag{16}$$

Intuitively, the terms $\overline{f_i}$ and $I_i(x, z|y)$ work together to enforce the representation $z$ to contain only the information needed to predict the label $y$, and no additional information (non-label-related) about $x$.

However, unfortunately, this mutual information term is not tractable because of $p_i(z|y)$ (which is hard to compute due to the integration over $x$). Therefore, we derive an upper bound in order to minimize it. Specifically:

**Lemma 1.** *Let $r(z|y)$ be a(ny) conditional distribution of $z$ given $y$ $\forall y \in \mathcal{Y}$. If $p(z|x)$ and $r(z|y)$ have the same support set $\forall x \in \mathcal{X}, y \in \mathcal{Y}$, we have:*

$$I_i(x, z|y) \leq \mathbb{E}_{p_i(x,y)}\left[\mathrm{KL}[p(z|x)|r(z|y)]\right] = \ell_i^{CMI} \tag{17}$$

*Proof.* Provided in the Supplementary Material. □

Here, the upper bound $\ell_i^{CMI}$ can be computed and used as a regularizer when training the representation network. We optimize both $p(z|x)$ and $r(z|y)$ in order to minimize $\ell_i^{CMI}$. Note that $r(z|y)$ can be a network that takes $y$ as an input and output the distribution $r(z|y)$; or more simply for classification (when we have a finite number of labels $y$), we can set $r(z|y) = \mathcal{N}(z; \mu_y, \sigma_y^2)$, where $\mu_y, \sigma_y^2$ ($y = \overline{1..C}$) are the parameters to be optimized, with $C$ being the number of classes.

Note also that if we use this (upper bound) regularizer $\ell_i^{CMI}$, it requires the representation mapping $p(z|x)$ to be probabilistic (due to the KL term). Hence, in the experimental section, we use a probabilistic representation network ($p(z|x) = \mathcal{N}(z; \mu_{w_1}(x), \sigma_{w_1}^2(x))$) for any models/variants with the $\ell_i^{CMI}$ objective, and use a deterministic network otherwise. With the empirical dataset $\mathcal{D}_i$ of client $i$, we have an empirical estimator of $\ell_i^{CMI}$ as below:

$$\ell_i^{CMI} = \mathbb{E}_{p_i(x,y)} \left[ \text{KL}[p(z|x)|r(z|y)] \right]$$

$$\approx \frac{1}{N_i} \sum_{n=1}^{N_i} \text{KL}[p(z|x_i^{(n)})|r(z|y_i^{(n)})] \tag{18}$$

**Connection to Domain Invariant Representation Learning:** We found out that this regularizer has a theoretical connection with conditional distribution alignment in domain generalization. We start with the following lemma:

**Lemma 2.**

$$\ell_i^{CMI} = \mathbb{E}_{p_i(x,y)} \left[ \text{KL}[p(z|x)|r(z|y)] \right] \geq \mathbb{E}_{p_i(y)} \left[ \text{KL}[p_i(z|y)|r(z|y)] \right] \tag{19}$$

*Proof.* Provided in the Supplementary Material. □

Therefore, by minimizing $\ell_i^{CMI}$, we also minimize the divergence $\text{KL}[p_i(z|y)|r(z|y)] \ \forall y$, forcing $p_i(z|y)$ and $r(z|y)$ to be close to each other: $p_i(z|y) \approx r(z|y)$. Therefore, the model will try to enforce $p_i(z|y) \approx p_j(z|y)(\approx r(z|y)) \ \forall$ clients/domains $i, j$, i.e., we are doing implicit alignment of the conditional distribution of the representation $z$ given the label $y$, which is a common and effective technique in conventional Domain Generalization. For example, [26, 25, 46] explicitly align this conditional distribution (not suitable for Federated Learning).

### 3.3.3 Implementation Choices and Optimization

For the federated optimization, we use FedAVG, which is one of the first and perhaps most common FL optimization algorithms (although we can use any federated optimization techniques).

Following common practice, we use mini-batches to approximate Eq. 8, Eq. 18, and Eq. 11 (instead of full-batch training).

For a probabilistic network (when using $\ell^{CMI}$), to allow the gradient to backpropagate through the samples $z_i^{(n)}$ to the network parameters $w_1$, we use the reparameterization trick [18]. With this reparameterization trick, $\overline{f_i}$ and $\ell_i^{L2R}$ can be implemented and back-propagated straightforwardly.

Regarding the conditional mutual information regularizer $\ell_i^{CMI}$, we use a Gaussian distribution for the variational conditional distribution $r(z|y)$, hence the KL term in Eq. 18 can be computed analytically (recall that we also use a Gaussian distribution for $p(z|x)$). Here the astute readers might wonder that such a simple choice of the variational distribution $r(z|y)$ might not be able to approximate $p_i(z|y)$. However, note that since both $p(z|x)$ and $r(z|y)$ are optimized in order to minimize $\ell_i^{CMI}$, it encourages the network to learn a simple representation so that the variational distribution $r(z|y)$ can match $p_i(z|y)$ (which arguably might be a good regularization effect). Furthermore, depending on datasets, we can, in theory, use more complex distributions such as a Gaussian mixture as the variational distribution. In such cases, one can use methods such as MC sampling to estimate the KL term. In our experiment section, for simplicity, we only use a Gaussian distribution for $r(z|y)$.

## 4 Experiments

### 4.1 Datasets

To evaluate our method, we perform experiments in four datasets (ranging from easy to more challenging) that are commonly used in the literature for domain generalization.

**RotatedMNIST [10]:** In this dataset, the MNIST images [19] are rotated counter-clockwise with an angle of $0°, 15°, 30°, 45°, 60°$ and $75°$ to form six domains $\mathcal{M}_0, \mathcal{M}_{15}, \mathcal{M}_{30}, \mathcal{M}_{45}, \mathcal{M}_{60}$ and $\mathcal{M}_{75}$. The task is classification with ten classes (digits 0 to 9). We use the same version of the dataset as in [31, 15] (where only 1000 images are rotated to form the domain) for easy comparison. Note that this is slightly different from the version in [11] and thus the numbers are not comparable.

**PACS [20]:** contains 9,991 images from four different domains: art painting, cartoon, photo, sketch. The task is classification with seven classes.

Table 1: **RotatedMNIST**. Reported numbers are from 3 runs

| | Models | Domains | | | | | | Average |
|---|---|---|---|---|---|---|---|---|
| | | $\mathcal{M}_0$ | $\mathcal{M}_{15}$ | $\mathcal{M}_{30}$ | $\mathcal{M}_{45}$ | $\mathcal{M}_{60}$ | $\mathcal{M}_{75}$ | |
| Centralized Methods | HIR [42] | 90.34 | 99.75 | 99.40 | 96.17 | 99.25 | 91.26 | 96.03 |
| | DIVA [15] | 93.5 | 99.3 | 99.1 | 99.2 | 99.3 | 93.0 | 97.2 |
| | DGER [46] | 90.09 | 99.24 | 99.27 | 99.31 | 99.45 | 90.81 | 96.36 |
| | DIRT-GAN [31] | 97.2 | 99.4 | 99.3 | 99.3 | 99.2 | 97.1 | **98.6** |
| FL methods | FedAVG [28] | 85.9±0.6 | 98.7±0.2 | 98.8±0.2 | 98.9±0.1 | 98.7±0.2 | 86.2±0.7 | 94.5 |
| | FedADG [44] | 89.9±0.5 | 98.9±0.2 | 99.0±0.1 | 99.1±0.1 | 98.9±0.2 | 90.2±0.4 | 96.0 |
| | FedCMI (ours) | 91.5±0.3 | 99.2±0.1 | 99.3±0.1 | 99.2±0.1 | 99.2±0.1 | 91.0±0.3 | 96.5 |
| | FedL2R (ours) | 90.5±0.5 | 99.1±0.2 | 99.2±0.1 | 99.1±0.1 | 99.2±0.1 | 90.7±0.6 | 96.3 |
| | FedSR (ours) | 91.6±0.3 | 99.3±0.1 | 99.3±0.1 | 99.2±0.1 | 99.3±0.1 | 91.5±0.3 | **96.7** |

Table 2: **PACS**. Reported numbers are from 3 runs

| | Models | Backbone | PACS | | | | Average |
|---|---|---|---|---|---|---|---|
| | | | A | C | P | S | |
| Centralized Methods | DGER [46] | Resnet18 | 80.70 | 76.40 | 96.65 | 71.77 | 81.38 |
| | DIRT-GAN [31] | Resnet18 | 82.56 | 76.37 | 95.65 | 79.89 | **83.62** |
| FL Methods | FedAVG [28] | Resnet18 | 77.8±0.5 | 72.8±0.4 | 91.9±0.5 | 78.8±0.3 | 80.3 |
| | FedADG [44] | Resnet18 | 77.8±0.5 | 74.7±0.4 | 92.9±0.3 | 79.5±0.4 | 81.2 |
| | FedCMI (ours) | Resnet18 | 80.8±0.4 | 73.7±0.2 | 92.8±0.5 | 79.5±0.2 | 81.7 |
| | FedL2R (ours) | Resnet18 | 82.2±0.4 | 75.8±0.3 | 92.8±0.4 | 81.6±0.1 | 83.1 |
| | FedSR (ours) | Resnet18 | 83.2±0.3 | 76.0±0.3 | 93.8±0.5 | 81.9±0.2 | **83.7** |

**OfficeHome [40]:** has 15,500 images of daily objects from four domains: art, clipart, product and real. There are 65 classes in this classification dataset.

**DomainNet [33]** is a large-scale dataset, consisting 586,575 images from 345 classes. These images are from 6 domains: clipart, infograph, painting, quickdraw, real, sketch.

## 4.2 Experimental Setting

For all datasets, we perform "leave-one-domain-out" experiments, where we choose one domain as the target domain, train the model on all remaining domains, and evaluate it on the chosen domain. Each source domain is treated as a client. Following standard practice, we use 90% of available data as training data and 10% as validation data. We train all our models with NVIDIA A100 GPUs from our AWS cluster.

For the RotatedMNIST dataset, we use a network of two 3x3 convolutional layers and a fully connected layer as the representation network $g_\theta$ to get a representation $z$ of 64 dimensions. A single linear layer is then used to map the representation $z$ to the ten output classes. This architecture is the same as the network used by [15, 31]. We train our network for 500 epochs with stochastic gradient descent (SGD), using a learning rate of 0.001 and minibatch size 64, and report performance on the test domain after the last epoch. Each client performs 5 local optimization iterations within each communication round ($E = 5$).

For the PACS datasets, for easy comparison with existing centralized domain invariant representation learning methods, we use the most common choice of backbone network in existing works as the representation networks, i.e., Resnet18 [14]. We use a ResNet50 [14] backbone for OfficeHome and DomainNet since they are more complex datasets. We replace the last fully connected layer of the backbone with a linear layer of dimension 512 for ResNet18 and 2048 for ResNet50 to form the representation network. As with the RotatedMNIST experiment, we use a single layer to map from the representation $z$ to the output. Each local client uses stochastic gradient descent (SGD) (a total of 5000 iterations) with learning rate 0.01, momentum 0.9, minibatch size 64, and weight decay $5e^{-4}$. Similar to the RotatedMNIST experiments, each client performs 5 local optimization iterations within each communication round ($E = 5$). However, we also found that our method is not sensitive to this

Table 3: **OfficeHome**. Reported numbers are from 3 runs

| Models | | Backbone | OfficeHome | | | | |
|---|---|---|---|---|---|---|---|
| | | | A | C | P | R | Average |
| Centralized | Mixup [43] | Resnet50 | 64.7 | 54.7 | 77.3 | 79.2 | **69.0** |
| Methods | CORAL [38] | Resnet50 | 64.4 | 55.3 | 76.7 | 77.9 | 68.6 |
| FL Methods | FedAVG [28] | Resnet50 | 62.2±0.9 | 55.6±0.9 | 75.7±0.2 | 78.2±0.2 | 67.9 |
| | FedADG [44] | Resnet50 | 63.2±0.9 | 57.0±0.2 | 76.0±0.1 | 77.7±0.5 | 68.4 |
| | FedCMI (ours) | Resnet50 | 61.8±0.5 | 55.5±0.9 | 76.3±0.1 | 77.4±0.1 | 67.8 |
| | FedL2R (ours) | Resnet50 | 64.5±0.3 | 56.5±0.5 | 76.1±0.2 | 77.9±0.2 | 68.8 |
| | FedSR (ours) | Resnet50 | 65.4±0.5 | 57.4±0.2 | 76.2±0.6 | 78.3±0.3 | **69.3** |

Table 4: **DomainNet**. Reported numbers are from 3 runs

| Models | | Backbone | DomainNet | | | | | | |
|---|---|---|---|---|---|---|---|---|---|
| | | | C | I | P | Q | R | S | AVG |
| Centralized | MLDG [21] | Resnet50 | 59.5 | 19.8 | 48.3 | 13.0 | 59.5 | 50.4 | **41.8** |
| Methods | CORAL [38] | Resnet50 | 58.7 | 20.9 | 47.3 | 13.6 | 60.2 | 50.2 | **41.8** |
| FL Methods | FedAVG [28] | Resnet50 | 59.3±0.7 | 16.5±0.9 | 44.2±0.7 | 10.8±1.8 | 57.2±0.8 | 49.8±0.4 | 39.6 |
| | FedADG [44] | Resnet50 | 60.9±0.6 | 17.2±0.2 | 44.3±0.2 | 12.4±0.2 | 57.6±0.9 | 50.3±0.8 | 40.4 |
| | FedCMI (ours) | Resnet50 | 59.0±0.9 | 18.0±0.7 | 44.6±0.5 | 12.2±0.4 | 56.2±0.2 | 50.0±0.4 | 40.0 |
| | FedL2R (ours) | Resnet50 | 60.2±0.6 | 18.1±0.4 | 44.9±0.6 | 11.0±0.9 | 57.8±0.4 | 51.5±0.7 | 40.6 |
| | FedSR (ours) | Resnet50 | 61.0±0.6 | 18.6±0.4 | 45.2±0.5 | 13.4±0.6 | 57.6±0.2 | 51.8±0.3 | **41.3** |

hyper-parameter (performance with $E = 100$ is almost identical). For the hyper-parameters of our method ($\beta_{CMI}$ and $\beta_{L2R}$), we follow [11] and use random search to tune their value based only on the validation set (no data leakage from the target domain). This protocol is highly recommended by [11] and allows for a fair comparison among the methods. For details of the tuned values, please refer to our supplementary material. Data augmentation is also standard practice for real-world computer vision datasets like PACS, OfficeHome and Domainet, and during the training we augment our data as follows: crops of random size and aspect ratio, resizing to 224 × 224 pixels, random horizontal flips, random color jitter, randomly converting the image tile to grayscale with 10% probability, and normalization using the ImageNet channel means and standard deviations. Our code will be released at https://github.com/atuannguyen/FedSR.

### 4.3 Baselines

For all experiments, we consider the FL algorithms to be our main baselines: FedAVG [28] (a common FL algorithm) and FedADG [44] (an existing FL algorithm with DG ability). For the RotatedMNIST and PACS experiments, we also include SOTA centralized domain-invariant methods for comparison (the numbers are taken directly from the original papers). For the OfficeHome and DomainNet experiments, since our setting (e.g., backbone network) and evaluation protocol are identical to those of DomainBed [11], our results can be compared directly to the results of the centralized methods recorded in that paper. Therefore, we also include the best performers for each dataset reported in the DomainBed paper. Note that we include these centralized methods for reference only, and they are not direct competitors of our model(s). For this reason, we highlight (in bold) the best performant model for the centralized and FL settings separately.

### 4.4 Results

Tables 1, 2, 3 and 4 show that our method (FedSR) outperforms the baseline FedAVG significantly (by a margin of 1.5% to 3% on average) and consistently in all experiments. FedADG [44] (a existing method for DG in the FL setting) does outperform FedAVG but is not yet on par with centralized methods. Meanwhile, FedSR achieves competitive results when compared to state-of-the-art centralized DG methods (as listed in the first half of the tables). This indicates the effectiveness of our method and the potential for Domain Generalization in the FL setting. As mentioned earlier,

our method can be interpreted as an implicit form of representation alignment. Please refer to our supplementary material for the visualization of our representation space, which does show the alignment effect of our method.

## 5 Conclusion

To conclude, in this paper, we present a simple and effective method for implicit alignment of the representation across domains/clients in a Federated Learning setting for a better domain generalization ability. We employ two regularization techniques, namely CMI and L2R, to learn a simple representation of the data in the hope for a better generalization. Furthermore, we also show that these regularizers implicitly align the marginal and conditional distribution of the representation, which are shown to be effective in the domain generalization problem. Extensive experiments show that our method outperforms relevant FL baselines (FedAVG, FedADG) and can even achieve competitive DG performance when compared to centralized approaches. A potential limitation of our method is that the invariance is enforced among the source domains and might fail to generalize to the unseen target domain (since the target domain can be arbitrary). However, this is a common comment for all domain invariant representation methods and not specific to ours. Furthermore, experiments show that our method does lead to better representation alignment and prediction performance in practice.

**Acknowledgments**    Author A. Tuan Nguyen acknowledges Meta AI for funding his PhD study. Oxford lab headed by author Philip Torr is supported by the UKRI grant: Turing AI Fellowship EP/W002981/1. Meta AI author Ser-Nam Lim is neither supported by the UKRI grant nor has any relationship to the grant.

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
