# OpenReview forum: "FedSR: A Simple and Effective Domain Generalization Method for Federated Learning"
_NeurIPS.cc/2022/Conference — NeurIPS 2022 Accept_

### Official Review · Reviewer_BRsg · 2022-07-03

**Rating:** 4
**Confidence:** 4
**Soundness:** 2 fair
**Presentation:** 2 fair
**Contribution:** 2 fair

**Summary:**

In the submission, a feature representation learning method was proposed to tackle the problem of federated learning based on the out-of-distribution classification setting. Specifically, two regularization terms were introduced, i.e., L2-norm and conditional mutual information regularization term. Experimental results are performed on 4 different benchmark datasets in the community of domain generalization, and some desired performances are reported.

**Questions:**

Followed by my first concern, since the main contribution is feature representation learning, therefore, it is necessary to discuss and compare with the DG baseline methods  which does not require domain alignment across domain, and can also be applied based on the FL setting.

**Limitations:**

The limitation of the proposed method has been discussed.

**Strengths And Weaknesses:**

Strength:

- The problem of federated learning based on OOD setting is interesting and sounding.
- The proposed method is simple and effective.

Weakness:

- While the proposed method is simple, IMHO, it does not shed light on federated learning. Noted that feature representation regularization is actually not novel in the community of DG (e.g., [A]),  these methods can also be applied to the individual domain followed by FedAvg.

[A] Adversarial Mutual Information-Guided Single Domain Generalization Network for Intelligent Fault Diagnosis


- FL based on heterogeneous environment (non iid) has been largely explored. It's interesting to see how these methods perform based on the setting DG.

- I suggest considering more realistic datasets (e.g., medical imaging), which require privacy guarantee for FL setting, for evaluation.

---

> ### Author Response · Authors · 2022-08-01
> **Response to reviewer BRsg**
>
> We would like to thank the reviewer for their time and effort in reviewing our paper. We address the reviewer's concerns below:
>
> **While the proposed method is simple, IMHO, it does not shed light on federated learning. Noted that feature representation regularization is actually not novel in the community of DG (e.g., [A]), these methods can also be applied to the individual domain followed by FedAvg.**
>
> The work (A) is indeed somewhat related to ours since they also use a Mutual Information term as a regularizer to tackle the single-domain domain generalization problem. However, we would like to remark that (A) needs to generate a "fake" target distribution from a source distribution and then maximize the mutual information between the representations of the two domains (source vs fake target). This might be overly complicated when applied at each client level in a FL setting and hence less practical.
>
> We emphasize that our method is **very simple** to implement, yet effective in this problem setting. Another major contribution of our work is also to make **theoretical connections** between the regularizer terms and the well-known representation alignment approaches in DG.
>
> Nonetheless, we will make sure to include the discussion of (A) in the revised version of our paper.
>
> **FL based on heterogeneous environment (non iid) has been largely explored. It's interesting to see how these methods perform based on the setting DG.**
>
> We have discussed this in our related work section (lines 74-79). We quote "although these above methods tackle the distribution shift across the source domains/clients, they still have not dealt with the generalization ability to an unseen target domain – the problem we consider in this paper."
>
> Since evaluating all these methods in our experiment setting is infeasible, we test Per-FedMe [1] (which is a recent and effective personalized FL method). This model indeed underperforms ours in our experiments. For example, it achieves 81.0% accuracy in PACS and 68.1% in OfficeHome. We can see that this model offers little (or no significant) improvement in the DG setting.
>
> [1] T Dinh, Canh, Nguyen Tran, and Josh Nguyen. "Personalized federated learning with moreau envelopes." Advances in Neural Information Processing Systems 33 (2020): 21394-21405.
>
>
> **I suggest considering more realistic datasets (e.g., medical imaging), which require privacy guarantee for FL setting, for evaluation.**
>
> Based on the reviewer's suggestion, we tried our best to conduct two additional experiments on large-scale and realistic datasets, namely the Camelyon dataset and FMoW dataset from WILDS [2] (each experiment is repeated 3 times). Camelyon is a medical imaging dataset as suggested by the reviewer, with 5 hospitals as 5 domains and the task is tumor prediction. Our method continues to show its effectiveness in these additional experiments.
>
> In this experiment, we use the same set of hyper-parameters as in the DomainNet experiment. The results are as follows:
>
> - Camelyon (Average Test ACC): FedAVG: 78.7%, FedADG: 79.4%, FedSR: 80.5%
> - FMoW (Average Test ACC): FedAVG: 51.2, FedADG: 53.0, FedSR: 54.1
>
> [2] Koh, Pang Wei, et al. "Wilds: A benchmark of in-the-wild distribution shifts." International Conference on Machine Learning. PMLR, 2021.
>
>
> **Followed by my first concern, since the main contribution is feature representation learning, therefore, it is necessary to discuss and compare with the DG baseline methods which does not require domain alignment across domain, and can also be applied based on the FL setting.**
>
> As discussed above, (A) might be overly complicated to be implemented in for each client in a FL setting. Most other non-representation alignment methods, for example, meta-learning based [3,4] or gradient-alignment based [5,6], also need access to multiple source domains in a centralized setting. More importantly, we show empirically that our performance is on par with even state-of-the-art centralized methods, so we believe our method will be competitive with any adaptation (if exists) of those methods to the FL setting (which is much more constrained than the centralized setting). In fact, FedADG can be viewed as an adaptation of DANN to FL, and our method outperforms this model consistently on multiple datasets.
>
> [3] Balaji, Yogesh, Swami Sankaranarayanan, and Rama Chellappa. "Metareg: Towards domain generalization using meta-regularization."(2018).
>
> [4] Li, Da, et al. "Learning to generalize: Meta-learning for domain generalization." 2018.
>
> [5] Parascandolo, Giambattista, et al. "Learning explanations that are hard to vary."  (2020).
>
> [6] Shi, Yuge, et al. "Gradient matching for domain generalization." (2021).
>
> Please let us know if you have in mind any specific DG model for the FL setting that you want to test!
>
> We thank the reviewer again for your time and hope that we have adequately addressed your concerns. We look forward to our discussion.
>
> Best regards,
>
> Authors.

---

> ### Author Response · Authors · 2022-08-07
> **A gentle reminder**
>
> Dear reviewer,
>
> We thank the reviewer for your valuable comments. As the reviewer is the only one leaning toward rejection of our paper (pre-rebuttal period), we especially spent more time and effort to address your concerns in our rebuttal. Since the authors-reviewers discussion period is ending in 2 days, we hope that the reviewer can spend some time going over our response to see if we have addressed your concerns and if you have any follow-up discussions/concerns/requests. If you are satisfied with our answers, please consider revaluating your rating of our paper. Otherwise, if you have any additional concerns, please do let us know before the discussion period ends!
>
> Best regards,
>
> Authors.

---

> > ### Comment · Reviewer_BRsg · 2022-08-08
> > **Feedback**
> >
> > Thanks for the response. My concerns about the evaluation have been addressed. However, I do not agree with the DG method we need to generate a fake target distribution. In the setting of DG, one can have multiple domains during training which can be used to simulate "covariate shift".  Again, I still don't find the intrinsic difference between the proposed method with the conventional DG without domain alignment, e.g., JigenDG [CVPR]. The contribution of only applying feature learning to FL setting is limited.

---

> > > ### Author Response · Authors · 2022-08-08
> > > **Continued discussion**
> > >
> > > Dear reviewer,
> > >
> > > Thank you for the prompt response. To clarify, we didn't mean that DG methods need to generate a fake target distribution. We were talking specifically about method [A] that the reviewer referred to, that it has to generate a fake target distribution and is overly complicated to be applied to a FL setting.
> > >
> > > It is indeed correct that "one can have multiple domains during training which can be used to simulate covariate shift", however, as we mentioned before, most of these methods use those source domains simultaneously and jointly together, thus not applicable to the FL setting where one client only has access to one domain.
> > >
> > > Thanks for referring us to JiGenDG, and we will include this baseline in the next revision of the paper, since this method can be applied straightforwardly to the FL setting. We note here that JiGenDG performs significantly worse than our method. We use directly the numbers reported by the authors: it achieves 80.51% in PACS and 61.2% in Office-Home, which are largely below ours (83.7 and 69.3). To be even fairer for JiGenDG, we re-evaluate this method in our setting (with the same processing code, backbone network, etc.), and got 80.4% and 67.1%, still significantly worse than our numbers.
> > >
> > > Note that JiGenDG's performance is not SOTA for DG anymore, since it doesn't explicitly exploit the interaction/relationship among the source domains (such as comparing the representation or the gradient). **The benefit of our method is that it can still exploit the cross-relationship among domains (representation alignment to learn an invariant representation), but does not require joint access to all the domains thus applicable to the FL setting**. Also, we have demonstrated in our experiments that our method performs as well as SOTA centralized DG methods. We quote again that **"we show empirically that our performance is on par with even state-of-the-art centralized methods, so we believe our method will be competitive with any adaptation (if exists) of those methods to the FL setting"**.
> > >
> > > Please let us know if you have any other baselines in mind for the FL setting that you want us to compare with. We promise to devote the rest of the discussion period to addressing this concern of yours.
> > >
> > > Best regards,
> > >
> > > Authors.

---

> > > > ### Author Response · Authors · 2022-08-09
> > > > **We wonder if you have any further comments.**
> > > >
> > > > Dear reviewer,
> > > >
> > > > Since the discussion period is almost over, we want to check again if we have addressed your concern about comparison and evaluation against non-alignment methods (that are directly applicable to the FL setting) adequately?
> > > >
> > > > We would like to reiterate that, methods that do not make use of the cross-domain information among the source domains (such as JiGenDG) are often less effective when dealing with domain shifts in the DG setting (their main benefit is only that they are readily applicable to the FL setting). The benefits of our method are that it is applicable to the FL setting but at the same time also aligns the representation among domains (cross-domain information), thus being more effective.
> > > >
> > > > We thank you again for your time!
> > > >
> > > > Best regards,
> > > >
> > > > Authors.

---

> > > > > ### Comment · Reviewer_BRsg · 2022-08-10
> > > > > **Thank you for more clarification**
> > > > >
> > > > > Thanks for more clarification. I do acknowledge the soundness of the proposed method. However, I am still not convinced how the proposed method benefits FL. There are plenty of methods focusing single domain generalization, which does not require domain interaction and can also be applied in FL setting, based on the claimant of authors. Also, there exists domain alignment method by aligning the feature distribution with a prior (such as Gaussian), which again does not require domain interaction. These methods are ignored by in the original manuscript and rebuttal. I am just curious, can all these method claim that there are designed for FL setting?
> > > > >
> > > > > Another important issue is the number of domains, as indicated by another reviewer, only using DG benchmark is very toy, and may not justify the real-world benefit to FL.
> > > > >
> > > > > To sum up, I keep my score unchanged.

---

### Official Review · Reviewer_FDaC · 2022-07-07

**Rating:** 6
**Confidence:** 4
**Soundness:** 3 good
**Presentation:** 3 good
**Contribution:** 2 fair

**Summary:**

This paper tackles the problem of domain generalization in the federated learning setting. Additional to optimizing the average of the local objective functions, this paper tries to learn the feature representation that is invariant with data distribution by optimizing a conditional mutual information $\ell^{CMI}$. An $\ell_2$-norm regularize is added to the loss function for the feature representation to regulate the feature complexity. The proposed algorithm is evaluated on several datasets to show its strength in dealing with data from unseen domains. The algorithm is compared with FedAvg and FedDGand a few centralized methods and has comparable performance as centralized algorithms.

**Questions:**

Q1: Could the authors include more discussions on existing works in domain generalization for federated learning and compare the proposed algorithm with the existing methods (e.g., [1.1], [1.2])

Q2: Can the author include an algorithm block demonstrating how $\ell^{CMI}_i$ is updated.

Q3: It is unclear how the data are separated as training/testing data. It is using one domain as testing data and the other domains as training data, or using 90% of the data from all domains as training and 10% of the data from all domains as testing data?

**Limitations:**

This paper has no negative social impact.

**Strengths And Weaknesses:**

Originality: This paper fails to include several important prior works in federated domain generalization, including [1,1], and [1,2], which are closely related to the proposed method in this paper. In specific, the idea of learning the feature representation with norm regularize has been used in [1.2]. The domain alignment (domain invariant representation) idea has also been used in [1.1]. Therefore, this paper seems to be incremental to the prior works.

Quality: The paper is well organized. It has theoretical justifications for the regularizers and made a clear connection to domain invariant representation learning. However, the analysis is still limited as no generalization or performance bound of the proposed loss is provided.

Clarity: The presentation of the paper is clear. The notations and concepts have been clearly defined and explained. Theoretical results are fully discussed and the numerical results are thoroughly explained.

Significance: This paper tackles an important issue in federated learning of the domain shift between the training and deployment phase. The numerical results show a promising improvement in different datasets.

[1.1] Peng, X., Huang, Z., Zhu, Y., & Saenko, K. (2019, September). Federated Adversarial Domain Adaptation. In International Conference on Learning Representations.

[1.2] Mohri, M., Sivek, G., & Suresh, A. T. (2019, May). Agnostic federated learning. In International Conference on Machine Learning (pp. 4615-4625). PMLR.

---

> ### Author Response · Authors · 2022-08-01
> **Response to reviewer FDaC**
>
> We thank the reviewer for helping with the reviewing process of our paper. We address your concerns and clarify some misunderstandings below:
>
> Q1: We would like to clarify the difference between our work and some of the existed works that the reviewer suggested as follow:
>
> **About [1.1]**: First of all, [1.1] considers a different setting than ours, e.g., federated domain adaptation: each client $i$ is still a source domain $D_i$, but there exists an additional target domain $D_t$ with an unlabeled dataset so that each source domain can align to. Secondly, each source domain in [1.1] aligns the representation distribution with the target domain by receiving the representations from this target distribution (and then performing adversarial training to pull the two representation distributions closer). This cannot be applied to our problem setting to align the representation of two clients, since one client cannot have access to another source client's data (including the data representation) for privacy reasons. Thirdly, FedADG (one of the baselines in our paper) can be viewed as an adaptation of [1.1] to our setting, since it uses the generator/discriminator framework to align the representation distribution of each source client to a reference (generated) distribution. As can be seen in the paper, our method outperforms this model by a large margin in all of the datasets, showing the effectiveness of our less complicated training objectives.
>
> We also would like to make a remark about the reviewer's comment that "our method lacks a theoretical bound on the model's loss in the target domain". While this kind of analysis is popular in the domain adaptation community (like [1.1]) since we have some information on the target domain; such a bound is often intractable for the domain generalization problem unless we make some very strong assumptions about the target domain (as the target domain can be "arbitrary") (for example, [1.2] assumes that the target domain is a mixture of the source domains, which is not true for our setting in both theory and in the experiments). Therefore, such an error bound is not common in the domain generalization community.
>
> **About [1.2]**: If we understand correctly, [1.2] uses the norm of the parameters as a regularization for the optimization. Our method, on the other hand, uses the L2-norm of the **representation**. Furthermore, we show how this is connected to marginal representation distribution alignment in the DG problem. Also note that we mentioned in the paper that these regularizers are common across multiple fields of machine learning. What we did was to revisit them and show their theoretical connections to representation alignment and their effectiveness in the DG experiments.
>
> We will include these discussions in the revised version of our paper.
>
> Q2: We have included a detailed description of how to compute $l^{CMI}$ in Appendix B.
>
> Q3: As mentioned in the paper (Section 4.2), we perform the "leave-one-domain-out" experiments, where we choose one domain as the target domain and the remaining domains as the source domains. Each source domain corresponds to a client. Each client (source domain) uses 90% of its data as the training set and 10% as validation data. After each round, the central server collects the validation accuracy from each client and average them to get the final validation accuracy. This evaluation protocol is recommended by the DomainBed codebase, which prevents any data leakage from the target domain to the validation set.
>
> We look forward to a fruitful discussion with the reviewer!
>
> Best regards,
>
> Authors.

---

### Official Review · Reviewer_dm5i · 2022-07-09

**Rating:** 7
**Confidence:** 4
**Soundness:** 4 excellent
**Presentation:** 4 excellent
**Contribution:** 3 good

**Summary:**

Motivated by the need for domain generalization in a federated learning setup (where each client has a different data distribution or domain), this paper proposes two simple regularizers based on the most common L2 norm squared regularizer and (an upper bound of) conditional mutual information (CMI), i.e., $I(x,z|y)$.
The L2 norm regularizer is shown to be related to minimizing the KL divergence between a fixed Gaussian distribution and the prior distributions at each client.
This can be seen as pushing all the prior distributions to the same *fixed* distribution.
Thus, this encourages marginal alignment of the latent distributions similar to AlignFlow.
The CMI regularizer is approximated via a variational approximation similar to the ELBO objective.
Because the model is shared, this encourages all client-specific conditional distributions $p_i(z|y)$ to match the global $r(z|y)$---thus making a connection to conditional alignment.
Finally, the paper gives empirical DG results on multiple datasets showing that both regularizers are helpful.


**Questions:**

- How is (11) related to the ELBO?  I expect this to be very close in terms of derivation using an auxiliary variational distribution r. Is the main difference that you are trying to match by conditioning on diferent things (i.e., $p$ conditions on $x$ but $r$ conditions on $y$)?

- Why discuss CMI first?  I would suggest putting L2R first.

- How does this relate to AlignFlow which aligns distributions by pushing them both towards a fixed standard normal distribution?

[AlignFlow] Grover, A., Chute, C., Shu, R., Cao, Z., & Ermon, S. (2020, April). Alignflow: Cycle consistent learning from multiple domains via normalizing flows. In Proceedings of the AAAI Conference on Artificial Intelligence (Vol. 34, No. 04, pp. 4028-4035).



**Limitations:**

- The paper mentions several limitations.


**Strengths And Weaknesses:**

Strengths:
- Proposes a simple algorithm that merely uses two locally-computable regularizers for domain generalization in a federated learning setting.
- Demonstrates insightful connections of these regularizers to marginal and conditional distribution alignment.
- Demonstrates the empirical effectiveness of these simple FL objectives for domain generalization on multiple datasets.
- Very easy to read paper that answers natural questions throughout the paper (e.g., Line 249-255, which I may suggest putting a little earlier or mentioning it a little earlier).

Weaknesses:
- It would be nice to evaluate these methods on real-world distribution shifts as in the WILDS benchmark dataset rather than only simulated or pseudo-realistic distribution shifts.

- An exploration of the number of optimization steps before synchronization (i.e., steps per round) would improve the paper by showing how this key FL parameter affects the results.  Currently, the synchronization is set at E = 5 (i.e., I think this means sync every 5 minibatches), which seems quite often unless this means 5 epochs. What is the performance if the models are synchronized every epoch, every 2 epochs or every half epoch?  Or what if E=10, 50, 100, 500?

[WILDS] Koh, P. W., Sagawa, S., Marklund, H., Xie, S. M., Zhang, M., Balsubramani, A., ... & Liang, P. (2021, July). Wilds: A benchmark of in-the-wild distribution shifts. In International Conference on Machine Learning (pp. 5637-5664). PMLR.

---

> ### Author Response · Authors · 2022-08-01
> **Response to reviewer dm5i**
>
> We thank the reviewer for their constructive feedback. We would like to answer the reviewer's questions as follows:
>
> **Additional Experiments**: Based on the reviewer's suggestion, we tried our best to conduct two additional experiments on large-scale and realistic datasets, namely the Camelyon dataset and FMoW dataset from WILDS (each experiment is repeated 3 times). We use the same hyper-parameter as in the DomainNet experiments. Below are the results:
>
> - Camelyon (Average Test ACC): FedAVG: 78.7%, FedADG: 79.4%, FedSR: 80.5%.
> - FMoW (Average Test ACC): FedAVG: 51.2, FedADG: 53.0, FedSR: 54.1.
>
>
> **Exploration of the number of local optimization steps**: As the reviewer suspected, in our experiments, each round consists of 5 local optimization steps (5 mini-batches). Since our paper focuses on "how to incorporate domain generalization to a decentralized learning setting" rather than other aspects of FL, we set the number of steps to 5 in all of our experiments and didn't explore in this direction much. However, as the reviewer suggested, we conducted an ablation study with E=10,50,100,500 with the PACS and DomainNet datasets. The result is that our method is not sensitive to the number of local optimization steps, as it achieves almost identical performance for E=5,10,50, and 100. For E=500, the model takes a (insignificant) hit of 0.6% on average accuracy in DomainNet, but so do other FL methods as we observe. Therefore, our model consistently outperforms other FL baselines in all settings. We conjecture that the performance hit (in the experiment with E=500 in DomainNet) can be further reduced with hyper-parameter tuning, but we think that these results are adequate to show that our method is not sensitive to the number of local optimization steps.
>
>
> **Differences between 11 and ELBO:** There are indeed similarities between our upper bound of the CMI and the evidence lower bound (for example, in a bayesian neural network or variational autoencoders), since both use a variational distribution to approximate an intractable distribution and form a lower (or upper) bound of the interested quantity (the CMI in our case and the evidence for ELBO). Additionally, equality holds when the variational distribution matches the distribution that is being approximated. These points also hold for essentially any type of variational approximation. Differences between our bound and ELBO are that we are trying to estimate (bound) a different quantity, and for a different purpose.
>
>
> **Moving L2R up** We have moved the discussion about L2R up to before that of CMI!
>
> **Connection to AlignFlow** Although both use the term "distribution alignment", they refer to different concepts. AlignFlow (and essentially normalizing flows) try to find an invertible function that map one distribution to another. The distributions are not "literally aligned" (as they are still distinct), just that we have a function to map back and forth between them. This has application in problems such as domain transfer (like StarGAN/CycleGAN). On the other hand, representation alignment is a common technique in Domain Generalization, as they pull the representation distributions of the domains closer (for example, by using some divergence/distance) so that the representation is invariant across domains and more likely to generalize.
>
> Please let us know if you have any follow-up questions!
>
> Best regards,
>
> Authors.

---

> > ### Comment · Reviewer_dm5i · 2022-08-06
> > **Thanks for the response and new experiments**
> >
> > Thank you for the responses. Overall, I appreciated the response and the new experiments.
> >
> > However, I would disagree that AlignFlow distributions are not "literally aligned".  If they are all projected to a Gaussian distribution in the latent space, then they are indeed aligned as they are all a standard normal.  The key difference is that in the MLE-only case of AlignFlow, the latent aligned distribution is fixed. However, for normalizing flows, if they are aligned in any space (e.g., the latent Gaussian), then the are also aligned in any invertible transformation of this latent space (e.g., because KL or JSD are invariant w.r.t. to invertible transformations applied to both distributions).  Additionally, AlignFlow also uses adversarially terms the encourage alignment in the observed space (and again any invertible transformation of this space is also aligned.  I don't think this is critical for the paper but I would encourage a deeper understanding of these papers.

---

> > > ### Author Response · Authors · 2022-08-07
> > > **Thanks**
> > >
> > > We would like to thank the reviewer for their comment. Indeed if we view alignflow as projecting two different distributions to a shared latent one, it is related to representation alignment as we want to project the input distribution to the same representation distribution. The differences would be that the projections in our case do not need to be invertible and the representation $z$ is only used for the prediction task.
> > >
> > > We thank the reviewer again for appreciating our paper and rebuttal!

---

### Official Review · Reviewer_AUa4 · 2022-07-11

**Rating:** 6
**Confidence:** 4
**Soundness:** 3 good
**Presentation:** 3 good
**Contribution:** 3 good

**Summary:**

This work proposes FedSR, a framework for learning (s)imple (r)epresentation in the (fed)erated setting so that the model can be applied to unseen domains/clients (in the sense of domain generalization). The representation is assumed to be one of the hidden layers in the neural network and is learned by minimizing the sum of (a) task prediction loss (2) conditional mutual information (CMI) and (3) L2 regularization (L2R) on the representation (see Eq.(9)). By theoretically justifying CMI and L2R terms, this work shows that together the objective function can lead to good generalization performance as demonstrated in common domain generalization benchmarks.

**Questions:**

The proposed FedSR provides a solution for domain generalization in the federated learning setting, where each client can only see data on their own data thus comparing data across clients is infeasible. The paper is well-organized and the presentation is clear. Each term in the objective function is well-motivated and their connections to common domain generalization techniques are also interesting. There are a few comments/questions

1. It may be better to explicitly mention that z is assumed to be conditional independent of y given x before Eq.(4)

2. For the experiment, are the hidden dimensions for z independent with separate variances? Or a covariance matrix is learned?

3. The experiments are only using a handful of clients (since each of them corresponds to one domain). However, it is common in federated learning to have tens to hundreds of clients, which the current experiments fail to demonstrate. In this sense, it would be more convincing if the clients are indeed sampled from an underlying distribution as depicted in Eq.(2) and (3). Then (2) and (3) will be important metrics for the baselines and FedSR.

4. It will be interesting to see what the representations of FedSR look like. Currently, the experiments only show the performance advantage in terms of accuracy, but we do not know why we have such an advantage. Maybe looking at the quality of representations can help, such as the CMI and L2R on the hold-out domain, or visualization of the representations.

Minor points

- LHS of (23) misses a p in the log
- L269 the variational distribution can match (what?)

**Limitations:**

The authors have sufficiently addressed the limitations.

**Strengths And Weaknesses:**

Strengths
- The problem is important yet under-explored in the literature
- The writing and presentation are clear and easy to follow
- Good empirical performance

Weaknesses
- The experiment setup is less typical for federated learning

---

> ### Author Response · Authors · 2022-07-31
> **Response to reviewer AUa4**
>
> We appreciate the reviewer's constructive comments about our paper. We clarify some of the details and answer the reviewer's questions below:
>
> - $z$ is indeed conditionally independent of $y$ given $x$, and we have newly added a sentence that clearly mentions this in Eq.(4) as suggested by the reviewer. However, note that this is not an assumption, it is rather a direct consequence of the fact that $z$ is a representation of (and only of) $x$ via the network $p(z|x)$.
>
> - The hidden dimensions of $z$ are independent with separate variances! Our distributions $p(z|x)$ and $r(z|y)$ are both diagonal Gaussian to save computational cost.
>
> - When we have plenty of test domains, the average/worst-case test performance as in Eq.(2) and (3) indeed will evaluate the methods more accurately. However, in the case of a limited number of domains as in most domain generalization datasets, it is a common practice in the DG literature to do the leave-one-domain-out experiments (each domain takes turns to be the target domain) and conduct the experiments across multiple datasets to confirm the effectiveness of a method. We follow this protocol in this paper.
>
> - We actually already presented a visualization of the representation of FedSR in the appendix. Please refer to appendix C.2 for the visualization and discussion. Also, the quantities $\ell^{L2R}$ and $\ell^{CMI}$ do not change much across domains (including the hold-out target domain) in all experiments, indicating some level of transferability of the quantities.
>
>
> Please let us know if this accurately addresses your concerns. We would like to discuss any additional questions that the reviewer may have.
>
> Best regards,
>
> Authors.

---

> > ### Comment · Reviewer_AUa4 · 2022-08-08
> > **Response**
> >
> > I want to thank the authors for the clarifications. I still think the number of clients is too small from the federated learning perspective. Thus I will keep my score.

---

> > > ### Author Response · Authors · 2022-08-09
> > > **Thank you**
> > >
> > > We thank the reviewer for your comment and suggestion.
> > >
> > > We agree that the regime of a large number of domains is worth exploring. We note that most existing DG methods focus on the setting where the number of domains is small (so that DG are most effective). A future work direction would be to systematically evaluate existing DG methods (including ours) in the setting of a large number of domains, and how effective/ineffective they are when we increase the number of domains.
> > >
> > > Best regards,
> > >
> > > Authors.

---

### Author Response · Authors · 2022-08-07
**The end of the discussion period is approaching.**

Dear reviewers,

As the end of the discussion phase is approaching, we wonder if you can spend some time going over our response and let us know if we have adequately addressed your concerns and if you have any additional ones (we thank reviewer dm5i for already having done so!). If you have any additional concerns/requests, please let us know by August 9th since we cannot actively discuss them with you after the discussion period.

We once again thank all the reviewers for your time reviewing and helping to improve our paper.

Best regards,

Authors.

---

### Public Comment · Authors · 2023-01-19
**Regarding main paper and appendix separation.**

Dear PC,

Sorry for this very late reply. Somehow I missed your email regarding the separation of main paper and appendix for the camera-ready revision (the email from Jan 11). I wonder if you can give me another chance to edit the manuscript.

Thank you for your consideration.

Best regards,
Tuan.

---

> ### Comment · Area_Chair_Jg24 · 2023-01-19
> **Reply**
>
> Please send the updated pdf to sherry@eventhosts.cc.
>
> Thanks,
>
> AC

---

### Meta-Review · Area_Chair_Jg24 · 2022-08-27

**Recommendation:** Accept
**Confidence:** Less certain

**Metareview:**

The paper studies the problem of domain generalization in federated learning and proposes a new regularizer, which is a combination of L2-norm regularizer on the representation and a conditional mutual information (between the representation and the data given the label) regularizer. The paper is well written and authors provide experimental results in domain generalization datasets.  Reviewers also raise several concerns about the paper. They remark that the experiments are not in the cross-device federated learning setup with many clients / domains,  the algorithm is not novel in the domain generalization community, lack of comparison with prior works. I strongly encourage authors to add more baselines from the domain generalization literature in the final version.

**Award:**

No

---

### Decision · Program_Chairs · 2022-09-14

Accept